# Quantification of Pulmonary Artery Configuration after the Arterial Switch Operation: A Pilot Study

**DOI:** 10.3390/diagnostics12112629

**Published:** 2022-10-30

**Authors:** Thomas Martens, Gillian Claeys, Joachim De Groote, Meletios Kanakis, Martin Kostolny, Victor Tsang, Marina Hughes

**Affiliations:** 1Department of Cardiac Surgery, Ghent University Hospital, Corneel Heymanslaan 10, 9000 Ghent, Belgium; 2Department of Pediatric and Congenital Heart Surgery, Onassis Cardiac Surgery Center, Kallithea, 17674 Athens, Greece; 3Department of Cardiothoracic Surgery, Great Ormond Street Hospital, London WC1N 3JH, UK; 4The Faculty of Medicine, Slovak Medical University, 83101 Bratislava, Slovakia; 5Department of Cardiology, Norfolk and Norwich University Hospital, Colney Ln, Norwich NR4 7UY, UK

**Keywords:** imaging, arterial switch operation, pulmonary artery, transposition of the great arteries

## Abstract

Background: The arterial switch operation (ASO) is the preferred treatment for d-transposition of the great arteries (TGA). Freedom from reintervention is mainly determined by the performance of the arterial outflow tracts, with variable incidence of pulmonary artery stenosis (PAS), possibly related to aspects of surgical technique. This pilot study attempts to describe pulmonary artery (PA) configuration through several measurements using three-dimensional data from cardiac magnetic resonance (CMR) imaging and assesses whether PA configuration is associated with PAS. Methods: A retrospective, single-centre analysis of paediatric patients undergoing CMR after ASO. The geometry of the pulmonary arteries was compared between patients with and without PAS as judged by the CMR report. Results: Among all patients (*n =* 612) after ASO, 45 patients underwent CMR at a median age of 10 years (3.5–13). Twenty-two (57.9%) had PAS, categorized as mild (*n =* 1), moderate (*n =* 19) or severe (*n =* 2). Eighteen had stenosis on PA branches. Four had MPA stenosis. Comparison between groups with and without PAS revealed no significant differences in neo-aortic to pulmonary angle, MPA to LPA/RPA angle, or bifurcation angle. There was a significant difference in cranial displacement, with more cranial displacement in the group without PAS. However, this group was older, 10.8 (7.3–14.3) years compared to those with PAS, 6.8 (1.5–12.1). Conclusions: The spectrum of PAS after ASO is heterogenous. This study shows the feasibility of measuring PA configuration in three planes on CMR. There is no correlation between PA configuration and PAS. Therefore, other mechanisms are probably responsible for the occurrence of PAS, rather than the configuration on itself. Further multicentric studies are warranted to confirm the suggested measuring method and assessing the associations with PAS, to eventually advise surgical methodology.

## 1. Introduction

The arterial switch operation (ASO) has been established as the preferred surgical treatment for simple d-transposition of the great arteries (TGA). Short-term outcomes are good, with an early mortality rate of 2.8% [1]. The freedom from reintervention and reoperation is mainly determined by the performance of the arterial outflow tracts. Neo-aortic root dilation and subsequent valve regurgitation remain a concern, with recent studies demonstrating this burden and documenting the progression of aortic root diameter, even during adulthood [2,3]. Additionally, there is a wide range of incidence of pulmonary artery stenosis (PAS), from 4% [4] to 80% [5] between different surgeons and institutions. The burden of resource needs for catheter or surgical intervention on PAS is significant, but the aetiology of PAS remains unclear.

As a part of the ASO, the French manoeuvre [6] creates a unique great vessel configuration, with the main pulmonary artery (MPA) positioned anterior to the aorta. From the anteriorly positioned pulmonary bifurcation, the branch pulmonary arteries course laterally to each side of the aorta. One hypothesis regarding the range of occurrence of PAS is that it is specifically related to elements of the original surgical technique and the subsequent geometry of the original repair. Understanding this has the potential to improve long-term outcomes of ASO.

PAS is usually detected through follow-up echocardiography, despite its recognized limitations, particularly in adult and obese patients [7]. More recently, the increasing availability of cardiac magnetic resonance (CMR) imaging has added substantial diagnostic sensitivity. The use of CMR for routine surveillance of ASO is suggested in several guidelines [8,9].

In an attempt to detect possible mechanisms for branch PAS, Morgan et al. proposed two measurements using CMR [10]. However, additional measurements in different planes could add valuable detail to the description of PA morphology, subsequently leading to better identification of mechanisms for PAS.

This study describes a pilot investigation using three-dimensional CMR data to make several systematic measurements of PA configuration in children after ASO. The relationship of these parameters to PAS was scrutinized to explore whether PA configuration itself is associated with PAS.

## 2. Methods

### 2.1. Overview

A pilot study was performed, through retrospective, single-centre, CMR analysis of paediatric patients who had undergone ASO for d-TGA with or without ventricular septal defect (VSD). The study was approved by the institution’s ethical committee, waiving the need for informed consent due to the retrospective nature of the study, which involved only in-house data review. Patients with complex disease, such as Taussig–Bing anomaly or additional aortic arch hypoplasia, were excluded.

### 2.2. Surgical Technique

All patients underwent standard arterial switch operation with coronary transfer, reconstruction of the PA using quadrangular fresh autologous pericardium and French manoeuvre. The PA reconstruction technique included mobilization of both pulmonary artery branches towards the hila, direct suturing of the pericardial patch to two-thirds of the annulus of the neo-pulmonary trunk and resuspension of the neo-pulmonary valve inside the patch. During the reconstruction, the length and configuration of the great vessels were visually checked by the operating surgeon. At the end of the procedure, after stopping extracorporeal circulation, the surgical result was confirmed through transoesophageal or epicardial echocardiography, focusing on cardiac function, valvular function and gradients over both outflow tracts.

### 2.3. Patient Cohort

All patients receiving an ASO between January 1997 and September 2017 at the Great Ormond Street Hospital in London were reviewed. For patients be included, the availability of at least one postoperative CMR was necessary. In the case of multiple available CMR’s, the most recent study was used for analysis.

Generally, there were two main groups of patients within this convenience sample of consecutive patients referred for CMR after ASO: patients of any age referred for investigation of suspected haemodynamic compromise, and asymptomatic teenage patients referred for routine, baseline assessment, prior to transfer of their care to adult congenital cardiac services.

The reported CMR data were evaluated, and the cohort was divided into two groups; those for whom the primary CMR report described PAS, and those for whom the primary CMR report did not describe any degree of PAS.

PAS was defined by the original CMR written report. The Radiologist interpretation of PAS was formed by composite analysis of BSA-indexed (Z-score) calibre of the neo-pulmonary valve, MPA, proximal LPA and proximal RPA, the flow velocity gradients recorded, and the proportion of flow to each lung.

For the purpose of this study, the PA was seen as an anatomical complex, including MPA and proximal PA branches. Stenosis at any level of this complex often carries comparable therapeutic repercussions, and delineation between MPA or PA branches sometimes is difficult, especially where they collide with each other, at the level of the bifurcation.

### 2.4. CMR Measurements

CMR was performed on a 1.5 Tesla scanner. Three-dimensional, respiratory navigated and ECG-gated steady-state free precession (SSFP) images were used for image analysis. These data were captured in the diastolic phase, and constructed with isotropic voxels. Spatial resolution (voxel size) ranged from 1.2 mm to 1.8 mm. Datasets were imported in Osirix (version 12.0.0, Pixmeo SARL, Geneva, Switzerland), a commercially available interface enabling visual evaluation and detailed analysis of DICOM files. The measurements were performed by two independent researchers (GC and JDG), after an initial comparative measurement analysis, which showed a good correlation between the observers. In 90%, the measurements did not differ more than 10%. Whenever the difference was > 10%, the values were judged by a third observer (TM).

### 2.5. Parameters (Figure 1)

Two geometric parameters, previously described by Morgan et al. [10], were used for this study:(1)Neo-pulmonary to neo-aortic angle

This is obtained in the transverse plane by measuring the angle between (A) and (B), where (A) = the axis from anterior to posterior through the centre of the neo-aortic root, and (B) = the axis through both centres of the neo-aortic and neo-pulmonary root (Figure 1, Panel A).

**Figure 1 diagnostics-12-02629-f001:**
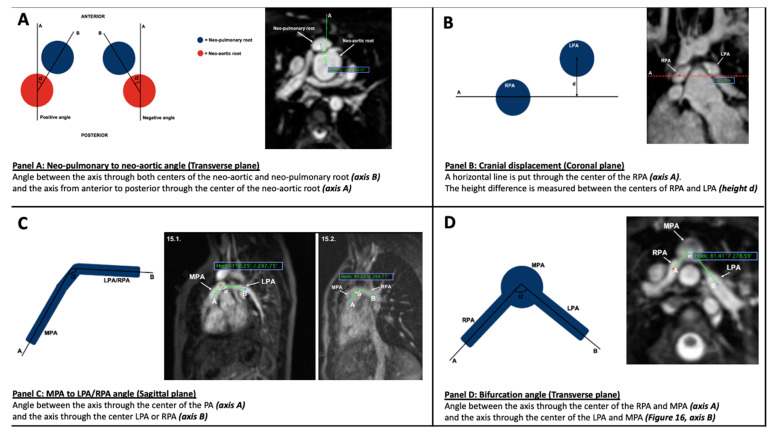
Presentation of measuring. Schematic diagram (left-hand side) with example on CMR image (right-hand side). Panel (**A**): Neo-pulmonary to neo-aortic angle Panel (**B**): Cranial displacement (mm). Panel (**C**): MPA to LPA angle and MPA to RPA angle (degrees). Panel (**D**): Bifurcation angle (degrees).

(2)Cranial displacement

This identifies the height difference between the left and right pulmonary arteries (LPA and RPA) in the coronal plane. This measurement is achieved by drawing a horizontal line through the centre of the RPA (axis A). The height difference is then measured between the centres of RPA and LPA (Figure 1, Panel B).

Additionally, for this Pilot Study, a third and fourth parameter were established:(3)MPA to left pulmonary artery (LPA) and right pulmonary artery (RPA) angle

This measurement is made in the sagittal plane. It measures the angle between the main axis of the MPA and the main axis of the RPA and LPA branches, respectively. This angle is influenced by the height of pulmonary artery anastomosis (Figure 1, Panel C).

(4)Bifurcation angle

This measurement is made in the transverse plane. The bifurcation angle is the angle between an axis through the centre of the MPA and RPA (axis A) and an axis through the centre of the MPA and LPA (axis B) (Figure 1 Panel D).

### 2.6. Statistical Analysis

Analysis was performed with SPSS version 26 (IBM, Armonk, NY, USA). Continuous variables are reported as median (range between 25th and 75th percentile) or mean ± SD depending on distribution.

Categorical variables are reported as counts and percentages. Univariate analysis was performed through Mann–Whitney U and the Student *t*-test depending on distribution. *p*-values < 0.05 were considered statistically significant.

## 3. Results

### 3.1. Patients

During the study period, 612 patients underwent the index ASO procedure (Figure 2). Among all patients after ASO, 45 patients underwent ≥ 1 CMR examination for various indications. Two patients with poor quality imaging (imaging data integrity impaired by movement artefacts) and two additional patients without French manoeuvre were excluded from analysis. Three patients had already undergone intervention for PAS and were removed from the cohort. This resulted in 38 patients available for clinical and CMR analysis. These patients were born between 2002 and 2016.

Twenty-four patients were male (63.2%). Thirty-seven patients (97.4%) had an atrial septal defect (Table 1). Twenty-eight patients (71.1%) underwent a balloon septostomy after birth. Almost half of the patients (*n =* 16 or 42.1%) had a ventricular septal defect. During the study period, five consultant congenital cardiac surgeons performed the ASO using a similar technique. The median age at the time of ASO was 11 days (7–22). The median age at the time of CMR was 10 years (3.5–13). Four patients were less than 1 year old at the time of CMR (4, 5, 8 and 10 months, respectively). A statistically significant difference was found in terms of age at the time of CMR between the group with and without PAS described in the original CMR report. The patients without PAS were older at the time of CMR.

#### CMR

Of all scanned patients, 21 (55.2%) had PAS, categorized in the radiology CMR report as moderate (*n =* 19) or severe (*n =* 2). Eighteen patients had stenosis on PA branches (RPA *n =* 2, LPA *n =* 5, both branches *n =* 10). Four patients had MPA stenosis. MPA stenosis with bilateral branch stenosis was seen in one patient.

Subsequent to the CMR assessment, of patients with PA stenosis, 5 underwent surgical correction and 4 endovascular treatment. The remaining 13 patients with PA stenosis had regular follow-up without intervention.

A graphic representation of CMR measurements is shown in Figure 3. The median neo-aorta to the pulmonary angle measured −14° (−30.5 to −5.8), median cranial displacement was 5.7 mm (3.8–8.0), median MPA to LPA angle was 103.5° (97.8–111), MPA to RPA angle was 99.5 (89.8–104.2) and median bifurcation angle was 87° (77.8–99.3). In all but one patient, the LPA was higher than the RPA.

Comparison between groups with and without PAS (Table 2) revealed no significant differences in terms of neo-aortic to pulmonary angle, MPA to LPA/RPA angle and bifurcation angle. There was a significant difference in cranial displacement, with more cranial displacement in the group without PAS. However, this group was older, with a mean age of 10.7 (SD 3.3) years in contrast to the group with PAS (mean age 6.49 - SD 5.4). This value was not indexed for body surface area and therefore this difference was expected and cannot be correlated to the presence of PAS.

## 4. Discussion

The ASO for TGA provides excellent long-term outcomes [1]. However, lifelong follow-up with advanced imaging is warranted, since aortic dilation and PAS are frequently reported complications [3,11,12,13] and can occur without overt symptoms.

The incidence of PAS varies widely, from 4 to 81% [5,10,14]. The exact aetiology has not yet been delineated and is probably multifactorial. Possible mechanisms include anastomotic fibrosis and geometric distortion or stretching of the PA branches around the dilated aorta [15].

From a technical standpoint, the transection of the great vessels, PA reconstruction, the French manoeuvre and subsequent reconnection create a unique configuration. Ideally, this results in laminar flow without energy loss from the MPA into both branches. We hypothesized that geometric PA configuration, which results from the original surgical technique, could be a contributing factor for PAS. Therefore, and in addition to two parameters suggested by Morgan and colleagues [10], we investigated measurements describing angles between MPA and PA branches, as well as between both branches (bifurcation angle).

For the purposes of this study, CMR was preferred for the analysis of PA configuration and PAS detection. Although echocardiography carries advantages (widespread availability, satisfactory resolution in smaller children, no need for sedation/anaesthesia and time consumption related to the investigation) and is the primary screening tool in congenital heart disease [16], limitations in detecting distal PA branches are disadvantageous for the evaluation of PA morphology. This was shown by Lang et al. [7], who demonstrated only one PA branch in 47% of all ASO patients older than 10 years. Additional benefits of CMR include the detection of myocardial scar by late gadolinium enhancement [17]. Therefore, CMR recently has been recommended in the follow-up after ASO [8,17], and a baseline surveillance CMR is suggested when the study can be tolerated without anaesthesia, generally around the age of eight years.

Our analysis shows that CMR was performed in 7% (45/612) of all ASO patients operated on over 20 years in a single high-volume institution. This could lead to selection bias in this study, and therefore, results here should be seen from the perspective of selected CMR examinations. Possible explanations for this relatively low number, are the availability, resource and time consumption of this investigation. Additionally, the need for general anaesthesia to allow detailed CMR in younger children may have increased the threshold for this investigation. In our institution, if there was a high pre-test probability, based on echocardiographic and clinical data, that percutaneous intervention was necessary and feasible, some patients were immediately referred for cardiac catheterization without prior CMR. Additionally, some patients may have moved out of the catchment zone serviced by our centre, and been followed elsewhere.

This pilot analysis only included a paediatric population, confirmed by the median age of 10 years at the time of CMR. During the study period, CMR was only a routine (asymptomatic) investigation for some teenagers, undergoing the transition to adult care. The main indication for CMR after ASO was the clinical or echocardiographic indication of haemodynamic compromise. In the present analysis, PAS was seen in more than half of patients undergoing CMR after ASO, and this is likely due to clinical selection bias. Comparing this number with existing literature is difficult, since the incidence varies widely, from 4 to 81% [5,10,14]. The differences may relate to different measuring modalities, and varying clinical thresholds. In our cohort, 22.7% of PAS patients subsequently underwent surgical correction and 18% were managed by endovascular treatment. The remainder of patients were under follow-up with no indication or decision for surgery or intervention.

Our results indicate the heterogenicity of PAS in terms of severity and localization. Severe stenosis was seen in only two cases and branches could be affected uni- or bilaterally. As previously reported [5], in only the minority of patients with PAS, additional invasive therapy was performed. We observed interventional (*n =* 4) or surgical (*n =* 5) treatment in 21 patients with PAS, comparable to other studies [18,19]. The decisions about pulmonary artery interventions after ASO are made cautiously as percutaneous interventions carry significant risks. Thorough evaluation of coronary arteries is mandatory as coronary re-implantation sites may be adjacent to sites of pulmonary artery stenosis. Potential stent implantation or stent re-dilation is contemplated and the risk of stent fracture and possible aortopulmonary fistula should be recognized [19]. CMR again is a reliable tool which should recognize the exact sites of PAS before any intervention.

In this study, we hypothesized that certain geometric configurations could be associated with PAS. In their effort to identify mechanisms for PAS, Morgan et al. [10] introduced a subset of two geometric parameters on CMR, the neo-pulmonary to neo-aortic angle in the transverse plane and cranial displacement in the coronal plane. We added two additional CMR parameters, the MPA to LPA/RPA angle in the sagittal plane and the bifurcation angle in the transverse plane. We found a good correlation between measurements performed by separate researchers. Ideally, this concept should be evaluated by a multi-centric and international cooperation initiative, preferably analysing a representative group of ASO patients at a standardized point after their initial procedure. Since routine surveillance CMR is currently advocated in every adult congenital heart disease patient at some point in their life, this could certainly be realized.

Analysis of the different parameters in this pilot study did not find a difference between patients with or without PAS. The reason for this is probably multifactorial. Firstly, the study cohort is small, limiting statistical power, and secondly, PAS is heterogenous in severity and localization. Our convenience cohort is from a single centre, where surgeons used similar surgical techniques. It is less likely to discover any influences of the length of MPA on the MPA to LPA/RPA angles. Therefore, the influence of this parameter in PAS could have been underestimated. Additionally, the influence of a dilated neo-aortic root is not evaluated in this study. The enlarged root creates a substrate for PA elongation or compression at the branch level. This was observed during this analysis and has been confirmed by other studies [10].

Importantly, the role of uniform branch PA hypoplasia resulting from decreased growth of the vessel has not been measured or included in this analysis. This would not be defined as focal “stenosis” in a typical CMR report but if unilateral would have a significant haemodynamic effect on flow distribution to each lung, and if bilateral would affect afterload for the RV and overall cardiovascular performance.

## 5. Conclusions

In conclusion, we can say that the measurement of the suggested parameters to describe PA configuration can be performed easily using commercially available and simple analysis interfaces. The spectrum of PAS after ASO is heterogenous, and treatment is only performed in a minority of cases. We could not find any correlation between PA configuration in itself and PAS in this small Pilot study, possibly suggesting other mechanisms for PAS, such as aortic root dilation. Further multi-centric studies are warranted to confirm the suggested measuring method and linking aortic diameters to PAS.

### Future Research

Future research in computational fluid dynamics and the evolution of software for cardiac imaging will certainly improve imaging of the pulmonary arteries after ASO. Three-dimensional volume reconstructed images by enhanced echocardiography, cardiac computed tomography and CMR may be used as an input for a virtual reality application which is going to help the diagnostic and potential surgical or interventional plan in these cases [20,21]. Four-dimensional flow imaging CMR is a promising diagnostic tool assessing many flow parameters and energetics along the pulmonary artery tree [22] and is likely to give valuable insight into the haemodynamic effects of geometric PA configuration.

## Figures and Tables

**Figure 2 diagnostics-12-02629-f002:**
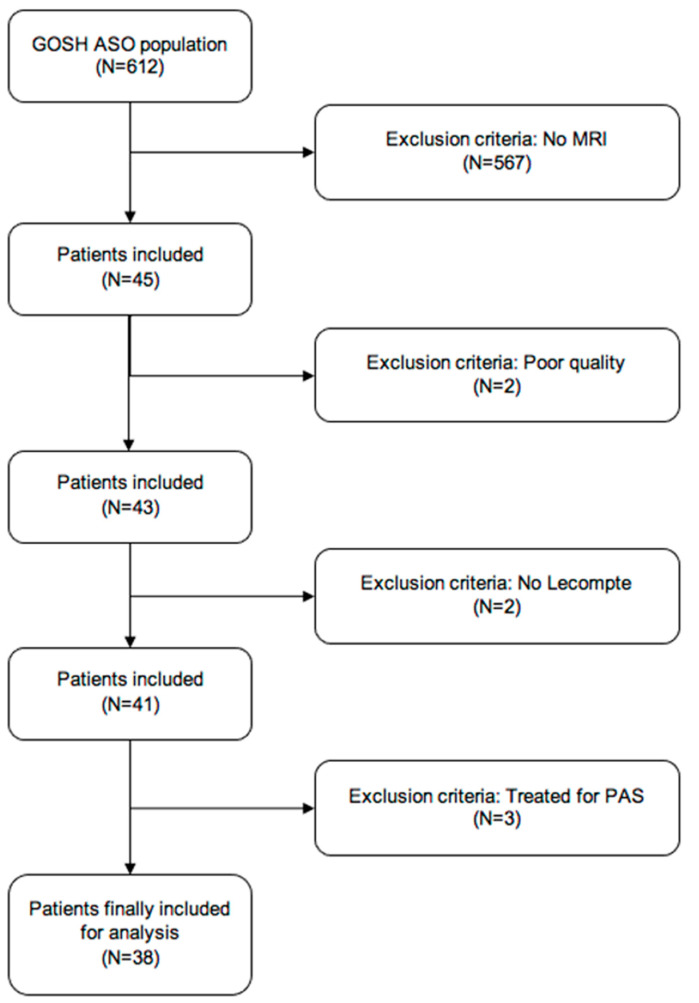
Patient selection.

**Figure 3 diagnostics-12-02629-f003:**
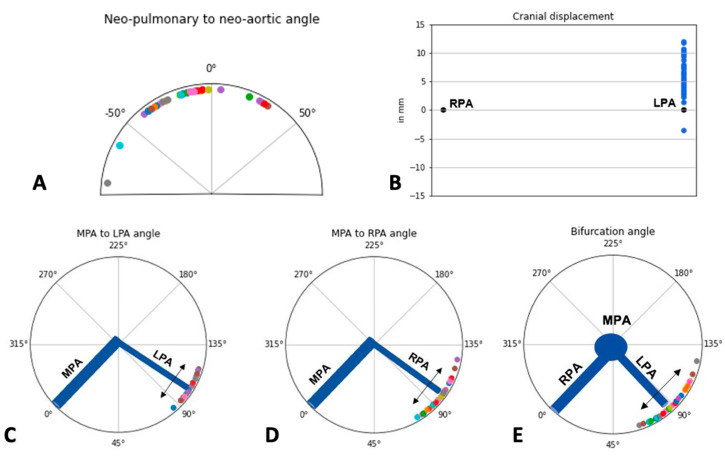
Graphic representation of CMR measurements for the different parameters. Panel (**A**): Neo-pulmonary to neo-aortic angle (degrees). Panel (**B**): Cranial Displacement (mm).Panel (**C**): MPA to LPA Angle (degrees). Panel (**D**): MPA to RPA Angle (degrees). Panel (**E**): Bifurcation Angle (degrees).

**Table 1 diagnostics-12-02629-t001:** Preoperative patient details.

	All *n =* 38 Median (IQR) or %	All *n =* 38 Mean (SD)	PAS *n =* 21 Mean (SD) or %	No PAS *n =* 17 Mean (SD) or %	*p*-Value
Age ASO (days)—SD	11 (7–22)	24 (47.6)	27.2 (42.4)	20.1 (18.2)	0.1
Age CMR (years)—SD	10 (3.5–13)	10 (5)	6.5 (5.4)	10.7 (3.3)	0.03
VSD (%)	42.1 (16)		43	41	
Septostomy (%)	73.7 (28)		67	82	

**Table 2 diagnostics-12-02629-t002:** Results of CMR measurements and comparison between PAS groups.

	Overall Mean (SD)	Overall Median (IQR)	Mean (SD) PAS Group *n =* 21	Mean (SD) Non-PAS Group *n =* 17	*p*-Value
Neo-pulmonary to Neo-aortic Angle (degrees)	−15.11 (23.77)	−14 (24.8)	−17.05 (22.39)	−12.44 (26.04)	0.747
Cranial Displacement (mm)	6.19 (2.84)	5.7 (4.2)	5.41 (2.72)	7.27 (2.72)	0.048
MPA-LPA (degrees)	104.03 (8.19)	103.5 (13.2)	103.23 (8.57)	105.13 (7.76)	0.473
MPA-RPA (degrees)	97.61 (11.19)	99.5 (14.5)	97 (11.71)	98.44 (10.76)	0.737
Bifurcation Angle (degrees)	88.87 (14.21)	77.8 (22)	89.45 (15.64)	88.06 (12.42)	0.759

## Data Availability

Additional data are available upon request.

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
