# Peer review of "Quantification of Pulmonary Artery Configuration after the Arterial Switch Operation: A Pilot Study"

_diagnostics, 2022, doi:10.3390/diagnostics12112629_

Round 1

Reviewer 1 Report

It was with great pleasure that I read the paper entitled “Quantification of pulmonary artery configuration after the arterial switch operation: a pilot study” . The topic of late sequelae after successful arterial switch after TGA is very up to date and relevant. Since the survival of patient with TGA is very good, patients with aortic dilatation, RVOTO and coronary problems are becoming the next challenge  in this patient population. 

The paper is original, and showed how with available imaging , some question regarding anatomic and surgical properties of PA configuration might be elucidated. The addition  and investigation of two additional geometric parameters is well appreciated.  

The paper is well written.   Limitation of the paper (e.g. volume, uniform surgical technique which might have underestimate some influential factors) is addressed. Furthermore the study is presented as a pilot study, suggesting that a follow up study will deal with these issue.

In conclusion I would support publication of this paper.

Author Response

Dear Reviewer, 

Thank you for your appreciation. It is true that this pilot study carries some limitations. We hope that this paper can create an initiative, scrutinizing PA configuration in a bigger ASO cohort, preferably multicentric. Thank you again for your positive review. 

Reviewer 2 Report

In this study the authors sought to evaluate the impact of   pulmonary artery  configuration through several measurements using 3-dimensional data CMR imaging on pulmonary artery stenosis after arterial switch. The topic is interesting and the parameters proposed could be interesting also.

 however, several limitations and comments have to be addressed

-          -Could the authors explain why the evaluate the pulmonary artery stenosis and pulmonary branches together, we can suppose that the geometry that can influence the stenosis of the branches and the main pulmonary artery could be different. Moreover the parameters proposed by Morgan et all where for pulmonary arteries branches stenosis if I’m not mistaken.

-        -  Did the authors test the geometrical parameters only for patients wit PA branches stenosis

-          -Selection bias, as the authors highlight the age affect the PS in their population, this could be explain by the indication to CMR in the younger population, did the authors test the result only in patient with elective indication or by correction of the analysis by age

-      -    Definition of pulmonary stenosis is not clearly defined, the authors reported  either pulmonary artery or pulmonary branches,the Radiologist interpretation of PAS was formed by composite analysis of BSA-indexed (Z-score) caliber of the neopulmonary valve, MPA, proximal LPA and proximal RPA, the flow velocity gradients, but none of those parameters are reported in the text

-         - Could the authors report pulmonary arteries parameters in the table for instance flow distribution, diameter or areas,etccc

Minor comments:

-          No patient have main PA and pulmonary branches stenosis?

Table 2  please add the number n,

MPA-LPA and others  please add angle and legend

Author Response

Thank you for your thorough review. We will address the comments step by step, and amend where necessary in the manuscript (marked in red). We believe these clarifications have improved the overall clarity of the work. 

-          -Could the authors explain why the evaluate the pulmonary artery stenosis and pulmonary branches together, we can suppose that the geometry that can influence the stenosis of the branches and the main pulmonary artery could be different. Moreover the parameters proposed by Morgan et all where for pulmonary arteries branches stenosis if I’m not mistaken.

Answer: Thank you for your question, it is an important one. The reason why we see the pulmonary artery complex (main pulmonary artery and proximal sidebranches) as one complex is fourfold:

  1. As we experienced during our analysis, in some cases different levels of this complex are affected by some degree of stenosis.
  2. The differentiation between distal PMA stenosis and proximal branch stenosis is sometimes difficult.
  3. There is no clinical repercussion for stenosis at these different levels (i.e. in case of severe MPA/branch stenosis, patients need endovascular or surgical therapy).
  4. Although a fair number of 38 studied patients, subdivision would have led to even smaller study sizes.

We highlighted this concept in the methods section (marked in red).

-        -  Did the authors test the geometrical parameters only for patients wit PA branches stenosis

Answer: We tested these parameters for patients with (n=21) and without (n=17) PA stenosis, as indicated in table 1.

-          -Selection bias, as the authors highlight the age affect the PS in their population, this could be explain by the indication to CMR in the younger population, did the authors test the result only in patient with elective indication or by correction of the analysis by age

Answer: Thank you for this question. Indeed, the patients that underwent ASO during the study period was 612 (as showed in Figure 2). Therefore, it is true that only a small amount of these patients underwent CMR for various reasons. Therefore, selection bias is certainly possible in this study. We added this selection bias in the discussion section.

-      -    Definition of pulmonary stenosis is not clearly defined, the authors reported  either pulmonary artery or pulmonary branches,the Radiologist interpretation of PAS was formed by composite analysis of BSA-indexed (Z-score) caliber of the neopulmonary valve, MPA, proximal LPA and proximal RPA, the flow velocity gradients, but none of those parameters are reported in the text

Answer: Thank you for this question. It is a valid one. We wanted to form two groups within the CMR ASO population, and subsequently analyze the PA complex configuration. We therefore needed the clinical entity of any PA stenosis, which naturally was based on the different radiological criteria used by our experienced cardiac radiologists (“The Radiologist interpretation of PAS was formed by composite analysis of BSA-indexed (Z-score) caliber of the neo-pulmonary valve, MPA, proximal LPA and proximal RPA, the flow velocity gradients recorded, and the proportion of flow to each lung”). As the main goal of the study was analysis of PA configuration (not done in the original radiology report),  we relied on the previous composite radiological end point of stenosis and did not re- examine whether the report was correct.  

-         - Could the authors report pulmonary arteries parameters in the table for instance flow distribution, diameter or areas,etccc

Answer: Please see the reponse on the last question. We did not gather these data for the purpose of this pilot study an relied on the expertise of our radiologists. But this is a valid remark, and we would encompass these parameters in a larger future multicentric study, where the definition of stenosis should be  better delineated since radiologists at different centers could have different measuring methods.  

Minor comments:

-          No patient have main PA and pulmonary branches stenosis?

 Answer: We again revised the data, and confirmed the majority of stenosis in both branches (n=10). Further stenosis were found in LPA (n=5), RPA (n=2), MPA (n=3), MPA and both branches (n=1). We added the latter this to the manuscript and omitted the case with mild stenosis.

Table 2  please add the number n, MPA-LPA and others  please add angle and legend

Answer: Done and highlighted in red. We also defined the units fort he different measurements.

We want to thank the reviewer again for thorough analysis and believe that these remarks have resulted in a better manuscript. 

Round 2

Reviewer 2 Report

The authors addressed he comments